# TOCOSH FLOUR (*Solanum tuberosum* L.): A Toxicological Assessment of Traditional Peruvian Fermented Potatoes

**DOI:** 10.3390/foods9060719

**Published:** 2020-06-02

**Authors:** Jonas Roberto Velasco-Chong, Oscar Herrera-Calderón, Juan Pedro Rojas-Armas, Renán Dilton Hañari-Quispe, Linder Figueroa-Salvador, Gilmar Peña-Rojas, Vidalina Andía-Ayme, Ricardo Ángel Yuli-Posadas, Andres F. Yepes-Perez, Cristian Aguilar

**Affiliations:** 1Academic Department of Pharmacology, Bromatology and Toxicology, Faculty of Pharmacy and Biochemistry, Universidad Nacional Mayor de San Marcos, Jr Puno 1002, Lima 15001, Peru; jvelasco9490@gmail.com; 2Department of Dynamic Sciences, Faculty of Medicine, Universidad Nacional Mayor de San Marcos, Av. Miguel Grau 755, Cercado de Lima 15001, Peru; jprojasarmas@yahoo.com; 3Clinical Pathology Laboratory, Faculty of Veterinary Medicine and Zootechnics, Universidad Nacional del Altiplano, Av Floral 1153, Puno 21001, Peru; rhanari@unap.edu.pe; 4School of Medicine, Faculty of Health Sciences, Universidad Peruana de Ciencias Aplicadas, Prolongación Primavera 2390, Lima 15023, Peru; pcmelfig@upc.edu.pe; 5Laboratory of Cellular and Molecular Biology, Biological Sciences Faculty, Universidad Nacional de San Cristóbal de Huamanga, Portal Independencia 57, Ayacucho 05003, Peru; gilmar.pena@unsch.edu.pe; 6Food Microbiology Laboratory, Biological Sciences Faculty, Universidad Nacional de San Cristóbal de Huamanga, Portal Independencia 57, Ayacucho 05003, Peru; vidalina.andia@unsch.edu.pe; 7Universidad Continental, Av San Carlos 1980, Huancayo 12000, Peru; ryuli@continental.edu.pe; 8Chemistry of Colombian Plants, Institute of Chemistry, Faculty of Exact and Natural Sciences, University of Antioquia-UdeA, Calle 70 52–21, A.A 1226, Medellin 050010, Colombia; andresf.yepes@udea.edu.co; 9Laboratory of Pathology, Instituto Nacional Cardiovascular, Jirón Coronel Zegarra 417, Jesús María 15072, Peru; a.crisaguilar@gmail.com

**Keywords:** *Solanum tuberosum* L., oral toxicity, tocosh, fermented foods, traditional medicine

## Abstract

Potato tocosh is a naturally processed potato for nutritional and curative purposes from traditional Peruvian medicine. The aim of this study was to investigate the acute and sub-acute toxicity of tocosh flour (TF). For sub-acute toxicity, TF was administered orally to rats daily once a day for 28 days at doses of 1000 mg/kg body weight (BW). Animals were observed for general behaviors, mortality, body weight variations, and histological analysis. At the end of treatment, relative organ weights, histopathology, hematological and biochemical parameters were analyzed. For acute toxicity, TF was administered orally to mice at doses of 2000 and 5000 mg/kg BW at a single dose in both sexes. Body weight, mortality, and clinical signs were observed for 14 days after treatment. The results of acute toxicity showed that the median lethal dose (LD_50_) value of TF is higher than 2000 g/kg BW but less than 5000 mg/Kg BW in mice. Death and toxicological symptoms were not found during the treatment. For sub-acute toxicity, we found that no-observed-adverse-effect levels (NOAEL) of TF in rats up to 1000 g/kg BW. There were statistically significant differences in body weight, and relative organ weight in the stomach and brain. No differences in hematological and biochemical parameters were observed when compared with the control group. For sub-acute toxicity, histopathological studies revealed minor abnormalities in liver and kidney tissues at doses of 5000 mg/Kg. Based on these results, TF is a traditional Peruvian medicine with high safety at up to 1000 mg/kg BW for 28 days in rats.

## 1. Introduction

*Solanum tuberosum* L. (Family: Solanaceae) is one of the most important Andean crops, cultivated along the Andean mountain range of South America and spread to other regions worldwide [1]. Over time, Andean farmers have developed frost and drought-resistant crops, which can be planted at heights greater than 3800 m above sea level (m.a.s.l.). In Peru, there are around 3800 varieties of potato and it is one of the main contributors to the world. The potato was domesticated just under 10,000 years ago; staple food crops of the ancient Peruvians not only used fresh potato but also consumed the product in the fermentation state, named tocosh [2].

Tocosh is a naturally processed potato for curative and nutritional purposes, which consists of leaving the potato in pools protected with straw or mesh near a stream for an average of six months, then it is extracted for consumption [3]. At the end of the process, the potato is reduced in size, except for its peel and it gets a very peculiar unpleasant smell. Since Inca and Pre-Incas times, the inhabitants of Ancash, Huánuco, and Junín regions have used tocosh as a medicine, in the form of flour or in its natural state to prepare *mazamorra* (Api in the Quechua language), is the best-known form of consumption (see Figure 1: Elaboration of tocosh). Tocosh flour (TF) is attributed to some beneficial properties such as gastritis, ulcers, gastro-esophagi reflux, and gastric cancer. People consume it by dissolving one teaspoonful per 100 mL in water before food as an alternative treatment. Although tocosh flour consumption is invariable, the normal dose known in traditional medicine is between 500 and 1000 mg/Kg daily (this information was taken according to an interview at the place where potato tocosh was collected).

Potato flour (PF) is characterized by its unpleasant smell, which is the first thing to be perceived, a peculiarity that does not limit its consumption or commercialization, affirming by empirical knowledge that it contains natural penicillin [4] and that among its innumerable benefits it is able to protect the gastric mucosa from damage or inflammation, according to popular customs, this product is used in postpartum, colds, pneumonia, in wound healing, as an antibacterial, healing of hemorrhoids and gastric ulcers, to avoid gastrointestinal infections and mountain sickness [5,6].

The potato species (*Solanum tuberosum* L.) are specific products consumed massively, these species present steroidal alkaloids, and when it is not well stored, can cause symptoms of poisoning such as respiratory distress, nausea, vomiting, and diarrhea related to acetylcholinesterase inhibition [7,8]. The primary steroidal glycoalkaloids in potato tubers are R-solanine and R-chaconine, being glycosylated forms of the steroidal alkaloid solanidine, these often improve the flavor of the potato [9]. The concentration of steroidal glycoalkaloids increases in response to several factors, such as injury, fungal attack, poor growing conditions, weather, and unsuitable storage conditions [10]. Nowadays, tocosh flour and derivates are sold as natural products in Peruvian markets but toxicity studies have not been reported to assess its consumption over a long period of time, which could induce any organ damage or death when there is no available data about the correct doses of administration. The main objective of this research was to evaluate the toxicological effect of tocosh flour following the guidelines for subacute and acute oral toxicity in rodents.

## 2. Material and Methods

### 2.1. Collection of Plant Material

Tocosh was collected in December 2019 from the Amarilis district, crossing the border of Yaca and Panao Pampa hamlets until reaching the Chicchuy village, Huanuco province (10°00′13″ S, 76°12′17″ W), Peru (see, Figure 1). Identification and authentication of the potato variety “walash” was used for elaborating tocosh and was carried out at the Natural History Museum of the Universidad Nacional Mayor de San Marcos (UNMSM), and a sample specimen was deposited with Ref. No. 038-USM-2020.

### 2.2. Preparation of the Tocosh Flour

The collected tocosh was washed in order to remove foreign matter and dust, then allowed to dry for three weeks under a shade in the place where it was obtained (−4 °C–18 °C). The dried tocosh was then pulverized using a grinder. The obtained product was named tocosh flour (TF) and stored until further use.

### 2.3. Phytochemical Analysis

A solution of tocosh dissolved and filtered was used to determine some phytochemicals such as alkaloids, phenols, terpenes, steroids, flavonoids, tannins, sugar, and saponins following the methodology of Herrera et al. [11]. The reaction to identify these components was done by using specific reagents for each chemical group showing any change of color or precipitation.

### 2.4. Experimental Animals

Balb/C albino mice and Holtzman rats of both sex were obtained from the bioterium of the National Institute of Health (Lima, Peru) with Sanitary Certificate No. 230-2019. Adult healthy male and female Holtzman albino rats (age, 12 weeks: body weight (males), 160–180 g; body weight (females), 150–170 g) were used to evaluate the sub-acute toxicity. Male and female rats were housed separately, and the selected female rats were nulliparous and non-pregnant. For acute toxicity, adult healthy male and female albino mice (age, 8 weeks: body weight (males), 30–32 g; body weight (females), 25–30 g) were used during the evaluation. These were kept in cages for approximately 15 days before the start of the study. The acclimatization of the experimental animals was carried out under environmental control conditions (12-h light/dark cycle) and temperature (22 ± 3 °C). Animals were given sterilized pellet food (National Center of Biological Products, NIH, Peru) and purified (reverse osmotic) water via a water bottle, ad libitum. All procedures were performed in reference to the institutional parameters and the guide for the care and use of laboratory animals. The protocol was presented and approved by the Ethics Committee of the Research Unit of the Faculty of Pharmacy and Biochemistry, UNMSM, (Document No 0198/FFB-UDI-2019. 11OCT2019. is certified with the REGISTRY No 010-CE-UDI-FFB-2).

### 2.5. Sub-Acute Toxicity at Repeated Dose for 28 Days

Sub-acute toxicity was performed in reference to the OECD 407 test guidelines [12]. Twenty rats were used and distributed in four groups. Group I (*n* = 5 males) and Group II (*n* = 5 females) named control groups, received only distilled water at repeated doses of 10 mL/Kg. Group III (*n* = 5 males) and IV (*n* = 5 females): rats received a limit dose of 1000 mg/Kg BW respectively, which according to animal safety criteria and empirical information for the consumption of tocosh by the population, which was established correspondingly to the repeated dose of 1000 mg/kg body weight, for 28 days. Each animal received a dose of tocosh flour suspension of 10 mL/kg body weight/day. The weights of each animal were recorded weekly during the treatment. After dosages of the product, the rats’ body weights were measured and recorded during the test every 7 days until completion at 28 days.

On the 29th day, blood samples were collected by intracardiac puncture, under anesthesia with ethyl ether, biochemical and hematological parameters were evaluated. At the same time, all the animals were sacrificed with sodium pentobarbital (100 mg/kg) by subcutaneous route. The organs were fixed in 10% formalin for histopathology examinations.

Organs such as the heart, lungs, liver, spleen, stomach, brain, kidneys, and testes or uterus were removed immediately after sacrifice, washed with 0.9% sodium chloride, dried on filter paper, and weighed calculating the relative weights of the organs (ratio of organ weight and animal body weight (at the end of the experiment) × 100). Organs were examined for gross and/or microscopic pathology.

The biochemistry exams were performed with the Liquid Kinetic Chemistry method by using a clinical chemical analyzer brand MEDICA—EasyRA (5 Oak Park Dr, Bedford, MA 01730, USA), according to the manufacturer’s specifications. The levels of aspartate aminotransferase (AST), alanine aminotransferase (ALT), alkaline phosphatase, total protein, bilirubin, total cholesterol, triglycerides, high-density lipoprotein (HDL), low-density lipoprotein (LDL), albumin, glucose, serum urea, and serum creatinine were determined.

Hematology examinations were performed by the flow cytometry method using a ZYBIO Brand Hematology Analyzer, Model Z31(Building J No. 70-1, 70-2 of Keyuan 4th Street Jiulongpo District, Chongqing Municipality 400039, China). The automated blood count (white blood cells, red blood cells, hematocrit, hemoglobin concentration, and platelet count) was evaluated.

### 2.6. Acute Oral Toxicity—Fixed-Dose Study Procedure

The acute oral toxicity of a fixed-dose procedure was evaluated according to the guidance of the Organization for Economic Cooperation and Development (OECD) method 420, with slight modifications in the animal selection, sex, and fasting [13]. This method grouped animals of both sexes dosing in a fixed-dose procedure using the highest doses of 2000 mg/kg and 5000 mg/kg (justified by criteria in animal welfare and related to the protection of human health based on the reference in the knowledge of empirical observation of the inhabitants of the area according to sample collection, being a product of frequent consumption in the area of the province of Huánuco).

Mice were kept with water ad libitum and were fasted for a fixed period such as 4–6 h before the administration of samples. Next, the animals were weighed and the test substance was administered. The toxicological evaluation was performed in 4 groups (*n* = 20). Group I (*n* = 5 males) and Group II (*n* = 5 females) received a single dose of 2000 mg/Kg body weight respectively, Group III (*n* = 5 males) and IV (*n* = 5 females): mice received a dose of 5000 mg/Kg BW, respectively. Each animal received a single dose of tocosh flour suspension at 10 mL/kg body weight. Mice were observed separately for 30 min, daily for 24 h, with rigorous observation in the first 4 h, and daily for 14 days. The individual weights of each mouse were determined before the administration of the test product and were re-calculated at the end of the 14 days.

The animal registry during the specific time of treatment was based on signs and symptoms of toxicity. The observations were recorded according to the duration of the treatment, including the specific external changes of minimal toxicity. At the end of the study, mice were sacrificed by sodium pentobarbital (100 mg/kg), immediately followed by necropsy. Organs were examined grossly for abnormal lesions.

### 2.7. Histopathological Analysis

Brain, heart, lung, liver, spleen. stomach, kidney, testes, and uterus of both studies (sub-acute and acute toxicity) were preserved in 10% formalin and fixed for a minimum of 24 h for a maximum of three days, dehydrated with alcohol of 70%, 96%, and absolute alcohol, the tissues were rinsed with xylol, impregnation with Paraffin, all these procedures were performed with a minimum of 1 h each. Inclusion in paraffin (formation of the paraffin block with the tissue) was performed, the cut was in a microtome, and at the last, they were sectioned at 3 µm depending on the tissue, and stained with hematoxylin and eosin (H&E). The sheets were assembled with coverslips and Entellan^®^ (Sigma Chemical Co, St. Louis, MO, USA), which is a rapid non-aqueous mounting medium. Finally, the organ slides were examined microscopically and photographed with an optical microscope Nikon Eclipse E200 (Shinagawa Intercity Tower C, 2-15-3, Konan, Minato-ku, Tokyo 108-6290, Japan) at 40× and 200× magnification [14].

### 2.8. Prediction of Drug-Likeness Properties for Steroidal Glycoalkaloids: α-Solanin, α-Chaconine and Solanidine

Drug-likeness prediction along with further ADME properties presents wide of opportunities to evaluate a rapid prediction of chemical compounds with possible toxicological effects. The drug-like and ADME properties for the most active components of tocosh flour (constituents chemicals from *Solanum tuberosum* L., Figure 2) were screened using open-access cheminformatics platforms such as Molinspiration (for molecular weight-MW, rotatable bonds, and polar surface area-PSA descriptors), ALOGPS 2.1 (for Log Po/w descriptor) and the Pre-ADMET 2.0 to predicted four pharmaceutical relevant properties such as intestinal permeability (App. Caco-2), albumin-binding proteins (KHSA), Madin-Darby Canine Kidney (MDCK Line) cells permeation and intestinal absorption (%HIA). These parameters establish movement, permeability, absorption, and action of potential drugs [15]. The interpretation of both MDCK and Caco-2 permeability using PreADMET is as follows:Permeability lower than 25: low permeability.Permeability between 25 and 500: medium permeability.Permeability higher 500: high permeability.

### 2.9. Statistical Analysis

Data were recorded indicating a mean ± SD of five animals in each group and were analyzed by Student *t*-test in Graph Pad Prism software v6. The results were significant when the *p* value is less than 0.05.

## 3. Results

### 3.1. Phytochemical Analysis

The phytochemicals present in the solution of tocosh flour were sugars, phenols, alkaloids, saponins, and steroids.

### 3.2. Repeated Dose Toxicity Study for 28 Days

All female and male rats that received tocosh flour at doses of 1000 mg/kg/day for 28 days survived and no signs of toxicity were observed. In relative organ weights, livers of both sexes had a significant increase (*p* < 0.05), which was compared with the control group. The other organs evaluated did not have a significant difference (*p* > 0.05) compared to the control group (Table 1).

There were no cases of death for 28 days. Microscopic examination of each organ in female and male rats showed no abnormalities due to toxicity in any of the organs, such as liver, kidney, heart, lung, stomach, brain, spleen, testes, and uterus, compared to the control group (Appendix A).

At the end of 28 days, liver enzyme levels were maintained relative to the control group in male and female rats, there was no variation in the respective biochemical parameters, only a slight variation in the significant increase in LDL cholesterol was recorded in rats and decreased triglycerides, in male rats. (Table 2). In hematological parameter, there was no variation except for a slight significant variation of the percentage of monocytes in male rats, the other results of the hematological parameters did not have significant differences with the control group (Table 3).

The body weight of the male rats administered with potato flour at a dose of 1000 mg/kg/day increased from the first week of 181.8 ± 9.28 g until the end of the fourth week, 261.4 ± 15.67 g, in contrast to the control group. In the control group of male rats, it was from 170.2 ± 9.20 g to 228.4 ± 20.24 g on day-28, which was significant (*p* < 0.05). Similarly, female rats administered at a dose of 1000 mg/kg had a weight gain that started in the first week at 172.8 ± 7.14 g to 228.4 ± 7.66 g corresponding to the end of the last week, compared to the control group of female rats which was significant (*p* < 0.001, see Figure 3).

### 3.3. Acute Oral Toxicity—Fixed Dose Procedure Study

The individual weights of the mice to which the tocosh flour was administered at doses of 2000 and 5000 mg/kg were determined before the administration of the test product and were calculated and re-recorded at the end of the 14 days, with weight gain at the end of the experiment.

Mice administered at doses of 2000 mg/Kg and 5000 mg/kg body weight exhibited minor organ damage in the liver (parenchymal lymphocyte) and kidney (lymphocyte) according to Figure 4 and Figure 5. However, no external aspects of toxicity were observed during the study linked to liver and kidney damage.

All this evaluation was followed in accordance with the OECD guideline 420, classifying the product as category B toxic (evident toxicity and/or ≤ 1 death), using the highest dose for safety and animal welfare to limit the number of animals, therefore the use of lower doses was restricted. In other organs, such as the brain, spleen, stomach, heart, and testes, no toxic damage was observed (Appendix A).

### 3.4. Prediction of Drug-Likeness Properties for Steroidal Glycoalkaloids: α-Solanin, α-Chaconine and Solanidine

We cannot show these results as toxicological finds due to these are only drug-likeness parameters. The properties showed the steroidal alkaloids as α-solanine, α-chaconine, and solanidine could be considered as pharmacokinetic behavior and were included because they are the main alkaloids found in *Solanum tuberosum* as is reported in other literature (Table 4).

## 4. Discussion

The glycoalkaloids from potatoes such as α-solanine, α-chaconine, and solanidine have a special function as a natural defense against plagues and its consumption may result in different symptoms that may include, diarrhea, fever, nausea, and even death [16,17]. The main studied mechanism of this kind of alkaloids is related to an inhibitory effect on the enzymes acetylcholinesterase (AChE) and butyrylcholinesterase (BuChE) [7,18]. Furthermore, it has been reported that glycoalkaloids interferes with ion-transport in cell membranes. The European Commission based on toxicological studies decided that the total glycoalkaloid content must not exceed the limit of 150 mg/Kg in potato protein powder for food applications [19]. However, the toxic effects produced by the inhibition of AchE and BuChE during the administration of tocosh flour were not observed in both subacute and acute toxicity treatments. Currently, there are no data on serious poisonings in the population, who consume tocosh flour as an alternative traditional treatment for digestive and respiratory diseases.

Tocosh comes from a fermentation process (Andean technique), which is suitable for distribution and consumption in the different markets of Peru as flour or in its raw form. On the other hand, the amount of glycoalkaloids are related to the cultivation method, storage and temperature, depending overall on the Andean techniques destined for its production and can be distributed in different rates in *Solanum tuberosum* tubers, they have been found in the tuber (smaller quantity), leaves and peel (greater quantity), and some analysis showed quantities such as 300–600 mg/kg in peel, 2000–4000 mg/kg in buds, and 3000–5000 mg/kg in flowers [20]. In our work, we did not evaluate the identification of constituents in tocosh flour by chromatography, but based on several studies of *Solanum tuberosum,* its glycoalkaloids are known by its fingerprint (α-solanine, α-chaconine and solanidine). Furthermore, probably other phytochemicals, lactobacillus and other could have been generated during the fermentation process and should be studied depending on the variety of potato and climatic factors. Otherwise, potato flour is an excellent food additive that can be applied in cakes, puffed food, breakfast food, baby food, condiments and soup and some functional factors such as anthocyanin, rosterone, and mucus proteins have been found, so its effect on health care is significant [21].

In regard to the repeated dose toxicity study using the limit dose of 1000 mg/Kg BW in rats, they had an increase in body weight. The increase of body weights differs from other studies where mice fed with 2130 mg/kg and 2170 mg/Kg of potato alkaloids such as α-chaconine and α-solanine for 7 days showed a decrease in body weight and in organs such as the liver, similarly in the administration for 14 days of the alkaloids: α-solasodine and solanidine [9,22]. This could be explained due to the animals being fed with tocosh flour which might contain a high content of carbohydrate and proteins, increasing the body weight in rats.

Significant increases in liver weight are commonly associated with adaptive changes such as accumulation of lipid, glycogen or other substances or a result of cell damage, congestion, hepatocellular hypertrophy or hyperplasia [23,24]. This variation does not always correlate with the amount of hepatic enzyme induction in rats as AST and ALT [25]. In our findings, we did not evidence any damage in liver tissues analyzed by using microscopy and any alteration in hepatic enzyme. This research revealed that presence of any phytochemical groups found in tocosh flour did not alter the histology in rats at the repeated dose of 1000 mg/kg BW for 28 days.

In hematological examination, there were no significant differences between rats administered with tocosh flour and the control group. The biochemical analysis LDL showed a significant increase in female and male rats. Similarly, there was a significant decrease of Tryglicerides in male rats but not in female rats. These findings might be altered due to potato starch, which is more phosphorylated than other starches of cereals [26], and indigestive polysaccharides promote excretion of bile acids, producing a reduction of Tryglicerides levels. However, an increase in LDL levels could be linked to carbohydrate consumption [27], thereby LDL production. No historical control data on the clinical chemistry or hematology values from the animal supplier were available for comparison; therefore, it is possible that the statistically changed parameters noted were within the normal range of average parameter values.

In acute toxicity according to OCED 420, minor changes in liver and kidney were observed with doses of 2000 and 5000 mg/Kg BW of tocosh administered in a single dose, probably attributed to its alkaloidal content and other components not determined in this study. *Solanum tuberosum* is known to have glycoalkaloids such as α-chaconin and α-solanine, mainly in tubers in almost 95%, also β-solanine, β-chaconin, γ-solanine, γ-chaconin, α-solamargin and β-solamargin but in less quantity [9,28]. However, these findings are unclear and not necessarily due to the consumption of tocosh flour.

The joint committee for food additives of FAO and WHO (JECFA) considers amounts of glycoalkaloids between 20 and 100 mg/kg as safe [29]. The toxic dose in the population corresponds to levels higher than 100 mg of total glycoalkaloids/kg of potato but this value could be influenced by environmental and storage conditions [17,30,31]. On the other hand, in the administration of 1.25 mg total glycoalkaloids/kg BW considered the highest dose in humans, gastrointestinal signs as vomiting appeared within 4 h [32,33]. Potato alkaloids at a dose of 75 mg/Kg of α-chaconin and α-solanin were lethally fatal within 4 to 5 days in Syrian golden hamster [34].

In an in vitro study, porcine oocytes were exposed to α-solanine (10 µM) and negatively affected early porcine embryo development by suppressing blastocyst formation and reducing embryo quality [35]. Another study found that the toxic and cytotoxic effects of α-Solanin in potatoes altered the proliferation and function of testicular cells in mice by regulating Sertoli and Leydig cells, affecting the testes and the reproductive function of male mice [36]. Likewise, α-solanine and α-chaconin at micromolar concentrations cause a cytotoxic effect on C6 rat glioma cells at the plasma membrane [37].

Currently, toxicological studies of the tocosh flour is not documented, according to the results obtained from in silico models, α-chaconine and α-solanine do not present Lipinski’s rules, this also results in the permeability values calculated using the Caco-2 and MDCK models being so poor (<25 nm/s). Additionally, the oral and intestinal absorption values are very low (<25%). However, it is important to highlight that the prediction values obtained for its ability to bind to plasma transporter proteins are within the range of 95%, so they would have a high toxicity when it is given by intravenous administration. Nevertheless, solanidine showed results for a quick absorption and might be due to its small molecular weight and chemical structure not glycosylated. In animal studies, it has been shown that orally administered glycoalkaloids are less toxic than intraperitoneal administration (I.P.) due to poor absorption in the gut. In mice, LD_50_ by I.P. administration have been reported to be 23 mg/Kg for α-chaconine, 34 mg/Kg for α-solanine, 500 mg/Kg for solanidine, and greater than 1000 mg/Kg for α-solanine by oral administration [38].

The NOAEL in rats of tocosh flour was 1000 mg/Kg, the estimate of the human equivalent dose (HED) [39] corresponds to a dose of 1000 mg as an initial dose in humans with 60 Kg of body weight. According to the traditional consumption of tocosh, this dose is less than the dose consumed by the population. Although, it seems to be safe compared with the limit dose of glycoalkaloids found in potato tubers.

In the present study, we could not identify and quantify each chemical constituent of tocosh potato. It is known that tocosh could contain other components such as lactobacillus and antibiotics produced during its fermentative process. However, tocosh can be consumed in established doses up to 1000 mg daily. Future studies of genotoxicity and chronic toxicity are needed as well to standardize its dose for consumption in foods.

## 5. Conclusions

The tocosh flour did not present toxicity at the repeated dose for 28 days in the highest dose corresponding to 1000 mg/kg BW. There were no deaths at up to 5000 mg/kg BW, therefore, the oral LD50 was greater than 5000 mg/kg.

## Figures and Tables

**Figure 1 foods-09-00719-f001:**
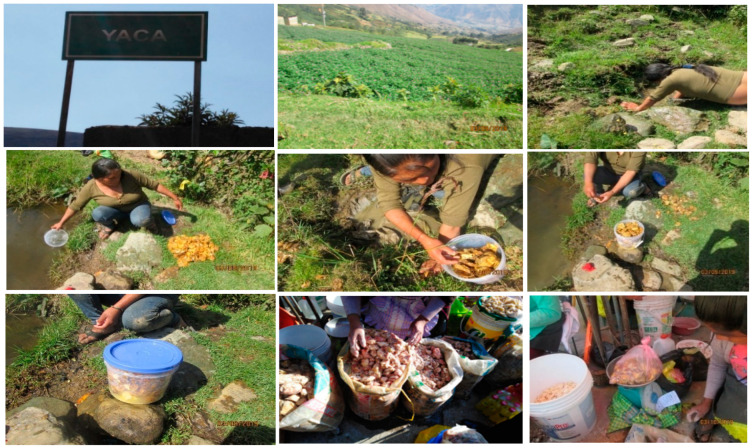
Extraction processing of “potato tocosh” in Yaca district, Huanuco region, Peru.

**Figure 2 foods-09-00719-f002:**
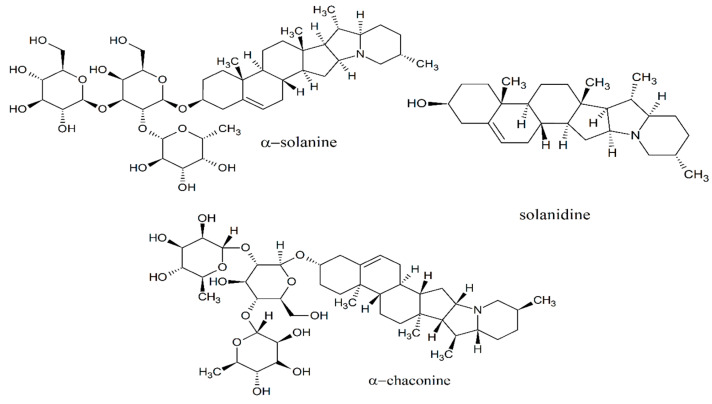
2D-structures for the main alkaloids found in potato tubers [10].

**Figure 3 foods-09-00719-f003:**
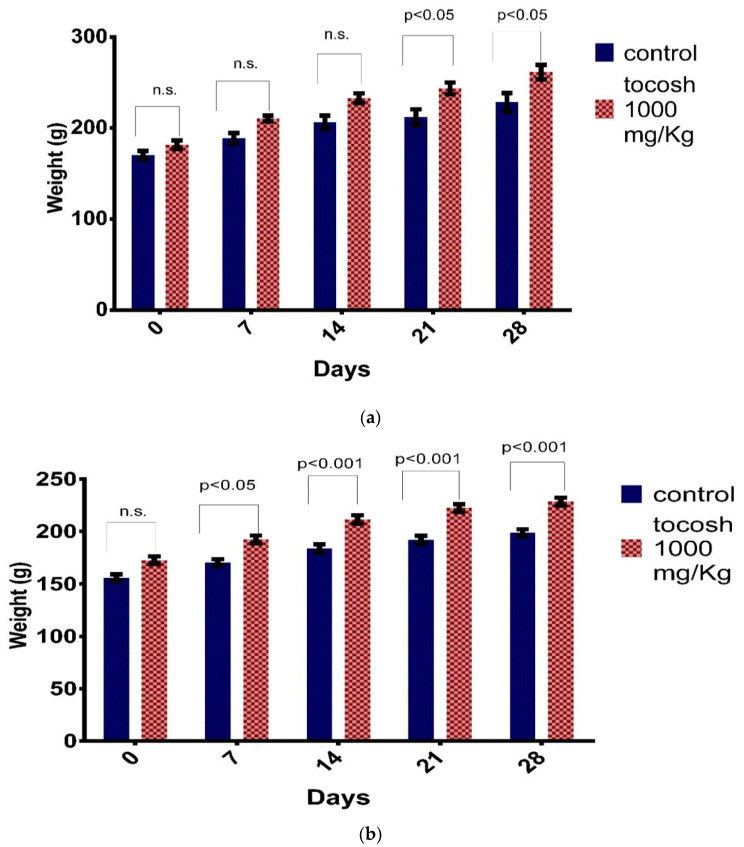
Body weights of rats treated with repeated oral doses of tocosh flour (1000 mg/kg) for 28 days. (**a**) male rats and (**b**) female rats. * *p* < 0.05, ** *p* < 0.001 compared to control, by using the Student *t*-test.

**Figure 4 foods-09-00719-f004:**
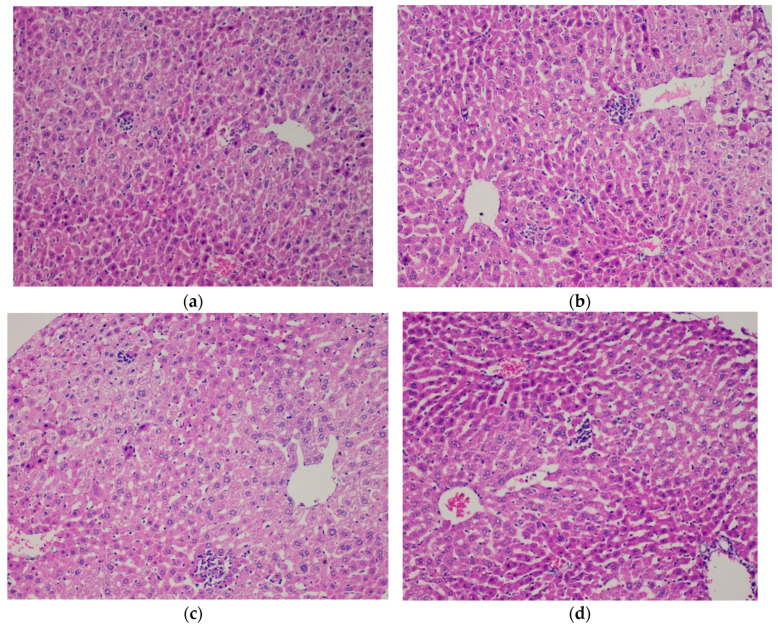
Liver tissue images of mice receiving a fixed dose of 2000 mg/kg and 5000 mg/kg of tocosh flour. (Stained with H&E, 200×). (**a**) Male group at the dose of 2000 mg/kg. Liver parenchyma with few lymphocytes around the central vein. (**b**) Female group at the 2000 mg/Kg dose. Liver parenchyma with isolated lobular lymphocytes (mild lobular inflammation). (**c**) Male group at the dose of 5000 mg/Kg. Liver parenchyma with isolated lobular lymphocytes (mild lobular inflammation). (**d**) Female group at the dose of 5000 mg/kg. Liver parenchyma with isolated lobular lymphocytes (mild lobular inflammation).

**Figure 5 foods-09-00719-f005:**
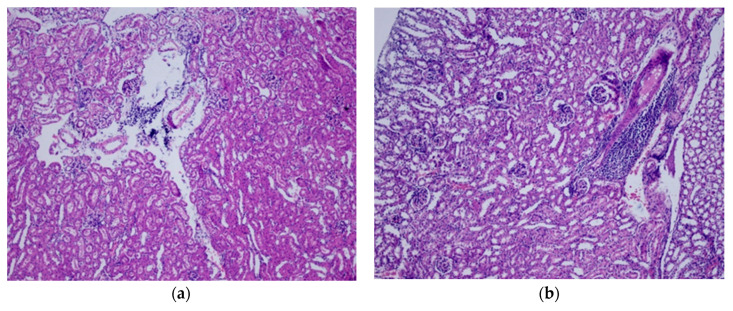
Renal tissue images of male and female mice that received a fixed dose of 2000 mg/kg BW and 5000 mg/kg BW of tocosh flour. (Stained with H&E, 200×). (**a**) Male group at the dose of 2000 mg/kg. Renal parenchyma with isolated chronic interstitial inflammatory infiltrate. (**b**) Female group at the dose of 2000 mg/kg. Renal parenchyma showing mild chronic interstitial inflammatory infiltrate with a tendency to lymphoid accumulation. (**c**) Male group at the dose of 5000 mg/kg. Renal parenchyma showing mild chronic interstitial inflammatory infiltrate with a tendency to the formation of lymphoid accumulation. (**d**) Female group at the dose of 5000 mg/kg. Renal parenchyma with isolated chronic interstitial inflammatory infiltrate.

**Table 1 foods-09-00719-t001:** Effect of tocosh flour solution on the relative organ weights in rats treated for 28 days.

Organs	Control Group	Tocosh Flour Dose: 1000 mg/kg
Male rats	*n* = 5	*n* = 5
Brain	0.70 ± 0.03	0.65 ± 0.01
Heart	0.35 ± 0.00	0.37 ± 0.05
Lung	0.48 ± 0.06	0.50 ± 0.15
Liver	2.96 ± 0.34	3.48 ± 0.33 *
Spleen	0.18 ± 0.04	0.26 ± 0.05
Stomach	0.76 ± 0.03	0.73 ± 0.06
Kidney	0.60 ± 0.07	0.76 ± 0.05
Testes	1.73 ± 0.05	1.77 ± 0.28
Female rats	*n* = 5	*n* = 5
Brain	0.78 ± 0.10	0.69 ± 0.05
Heart	0.40 ± 0.01	0.41 ± 0.03
Lung	0.47 ± 0.12	0.62 ± 0.04
Liver	2.83 ± 0.02	3.28 ± 0.25 *
Spleen	0.24 ± 0.04	0.26 ± 0.04
Stomach	0.62 ± 0.01	0.54 ± 0.04
Kidney	0.66 ± 0.03	0.68 ± 0.01
Uterus	0.47 ± 0.00	0.45 ± 0.06

Values are expressed as mean ± SD, compared with the control is significant when * (*p* < 0.05). Data were analyzed using the Student *t*-test. No historical control values are available.

**Table 2 foods-09-00719-t002:** Biochemical parameters of rats after administration of repeated oral doses of tocosh flour at doses of 1000 mg/kg for 28 days.

Parameters	Control Group	Tocosh Flour Dose: 1000 mg/kg
Male rats	*n* = 5	*n* = 5
AST (IU/L)	113.00 ± 4.00	108.50 ± 1.50
ALT (IU/L)	69.50 ± 3.50	70.00 ± 1.50
Alkaline Phosphatase (IU/L)	231.00 ± 9.00	238.50 ± 0.50
Total Bilirubin (mg/dL)	0.10 ± 0.00	0.10 ± 0.00
Total Protein (g/dL)	6.95 ± 0.05	7.01 ± 0.05
Albumin (g/dL)	3.80 ± 0.20	3.95 ± 0.05
Globulin (g/dL)	3.15 ± 0.25	3.00 ± 0.10
Total cholesterol (mg/dL)	59.50 ± 0.50	60.00 ± 0.00
Triglycerides (mg/dL)	134.50 ± 8.50	107.00 ± 13.00 *
HDL (mg/dL)	12.50 ± 0.50	10.60 ± 0.40
LDL (mg/dL)	20.00 ± 3.00	28.00 ± 3.00 *
Glucose (mg/dL)	100.50 ± 0.50	108.50 ± 1.50
Serum urea (mg/dL)	37.50 ± 1.50	40.50 ± 2.50
Serum creatinine (mg/dL)	0.44 ± 0.01	0.44 ± 0.03
Female rats	*n* = 5	*n* = 5
AST (IU/L)	116.50 ± 0.50	106.00 ± 0.50
ALT (IU/L)	65.50 ± 0.50	63.00 ± 1.00
Alkaline Phosphatase (IU/L)	235.50 ± 5.50	223.00 ± 17.00
Total Bilirubin (mg/dL)	0.10 ± 0.00	0.10 ± 0.00
Total Protein (g/dL)	6.90 ± 0.30	6.90 ± 0.20
Albumin (g/dL)	3.80 ± 0.20	4.10 ± 0.00
Globulin (g/dL)	3.05 ± 0.05	2.80 ± 0.20
Total cholesterol (mg/dL)	57.00 ± 3.00	59.50 ± 0.50
Triglycerides (mg/dL)	139.00 ± 1.00	130.50 ± 10.50
HDL (mg/dL)	9.55 ± 0.25	8.25 ± 0.25
LDL (mg/dL)	18.00 ± 2.00	25.00 ± 2.00 *
Glucose (mg/dL)	110.50 ± 3.50	113.00 ± 2.00
Serum urea (mg/dL)	35.50 ± 2.50	34.00 ± 0.00
Serum creatinine (mg/dL)	0.40 ± 0.02	0.46 ± 0.01

Values are expressed as mean ± SD, compared with the control is significant when * (*p* < 0.05). Data were analyzed using the Student test. AST (Aspartate aminotransferase); ALT (Alanine Aminotransferase); HDL (high density lipoprotein); LDL (low density lipoprotein). No historical control values are available.

**Table 3 foods-09-00719-t003:** Hematological evaluation of rats after administration of repeated oral doses of tocosh flour at doses of 1000 mg/kg for 28 days.

Parameters	Control Group	Tocosh Flour Dose: 1000 mg/kg
Male rats	*n* = 5	*n* = 5
RBC (×10^6^/mm^3^)	7.09 ± 0.30	7.30 ± 0.08
White blood cells (×10^3^/mm^3^)	4.14 ± 0.13	3.91 ± 0.17
Hemoglobin (g/dL)	14.70 ± 0.60	14.65 ± 0.15
Hematocrit (%)	41.80 ± 1.80	43.55 ± 1.45
Eosinophils (%)	1.50 ± 0.50	1.00 ± 0.00
Basophil (%)	0.00 ± 0.00	0.00 ± 0.00
Monocytes (%)	1.00 ± 0.00	2.00 ± 0.00 *
Segmented (%)	20.50 ± 8.50	17.00 ± 4.00
Lymphocytes (%)	77.00 ± 9.00	80.50 ± 4.50
Platelets (×10^3^/mm^3^)	7.5 ± 0.26	7.22 ± 0.22
Female rats	*n* = 5	*n* = 5
RBC (×10^6^/mm^3^)	7.00 ± 0.10	6.93 ± 0.22
White blood cells (×10^3^/mm^3^)	4.03 ± 0.18	4.20 ± 0.30
Hemoglobin (g/dL)	14.65 ± 0.25	14.60 ± 0.10
Hematocrit (%)	42.00 ± 0.00	41.50 ± 0.50
Eosinophils (%)	0.00 ± 0.00	0.00 ± 0.00
Basophil (%)	0.00 ± 0.00	0.00 ± 0.00
Monocytes (%)	2.50 ± 0.50	1.50 ± 0.50 *
Segmented (%)	24.00 ± 1.00	17.00 ± 1.00
Lymphocytes (%)	79.00 ± 1.00	80.50 ± 0.50
Platelets (×10^3^/mm^3^)	7.3 ± 0.20	7.6 ± 0.35

Values are expressed as mean ± SD, compared with the control is significant when * (*p* < 0.05). Data were analyzed using the Student test. No historical control values are available.

**Table 4 foods-09-00719-t004:** Calculated drug-likeness properties for glycoalkaloids α-solanine, α-chaconine, and solanidine.

Compound	M.W. ^a^	PSA ^b^	n-Rot Bond (0–10)	n-ON (<10) ^c^	n-OHNH ^d^	Log *P_o/w_* ^e^	LogK_HSA_ ^f^	Caco-2 ^g^ (nm/s)	App.MDCK (nm/s) ^h^	% HIA ^i^	% HOA ^j^	Lipinski Rule of Five
α-chaconine	852.070	199.560	15	15	8	−0.091	−0.875	<25 poor	<25 poor	<25poor	Low	3
α-solanine	868.069	220.534	17	16	9	−0.860	−1.157	<25 poor	<25 poor	<25poor	Low	3
Solanidine	397.643	22.288	1	2	1	5.007	0.554	1049	576	100	High	0

^a^ Molecular weight of the hybrid (150–500). ^b^ Polar surface area (PSA) (7.0–200 Å^2^). ^c^ n-ON number of hydrogen bond acceptors ≤10. ^d^ n-OHNH number of hydrogens bonds donors ≤5. ^e^ Octanol water partition coefficient (log *P_o/w_*) (−2.0 to 6.5). ^f^ Binding-serum albumin (K_HSA_) (−1.5 to 1.5). ^g^ Human intestinal permeation (<25 poor, >500 great). ^h^ Madin-Darby canine kidney (MDCK) cells permeation. ^i^ Human intestinal absorption (% HIA) (>80% is high, <25% is poor). ^j^ Model for Human Oral Absorption.

## Data Availability

The data sets used and/or analyzed during the current study are available from the corresponding author upon reasonable request.

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
