# Peer review of "TOCOSH FLOUR (*Solanum tuberosum* L.): A Toxicological Assessment of Traditional Peruvian Fermented Potatoes"

_foods, 2020, doi:10.3390/foods9060719_

Round 1

Reviewer 1 Report

The authors conduct acute and subacute toxicity study of TF, traditional Peruvian fermented potatoes. Also they state the test material is safe and not toxic. The intention of this study, and experimental methods are acceptable, however, to evaluate exactly a toxicity of the extract and assess a risk for human use as dietary product, a long-term study data conducted in rodents for 13 week should be included in this manuscript. 

Major comments

  1. A toxicity information in 13 week rat study with GLP standard should be included in the manuscript.
  2. There is lack of some information (rationale of animal used, diet information, 3 dose levels..)
  3. Out of normal range in hematological parameters such as hematocrit and hemoglobin

Author Response

Dear Reviewer:

Thank you for your suggestions with our paper. Additional changes were highligted with yellow and purple color.

Major comments

1. A toxicity information in 13 week rat study with GLP standard should be included in the manuscript.

R1: Thank you for your suggestions, We did not design a subchronic study for tocosh flour, only a subacute and acute study because this natural product is consumed for 15 days in the traditional peruvian medicine at the maximum time. People with gastritis generally find a beneficial effect in a short time as it is stated in the introduction of this manuscript and referenced. 

(4) Mayta-Tovalino F, Sedano-Balbin G, Romero-Tapia P, et al. Development of New Experimental Dentifrice of Peruvian Solanum tuberosum (Tocosh) Fermented by Water Stress: Antibacterial and Cytotoxic Activity. J Contemp Dent Pract. 2019;20(10):1206–1211.

However, we mention a recommendation to design a subchronic study in the future in discussion section.

2. There is lack of some information (rationale of animal used, diet information, 3 dose levels..)

R3: Thank you for your suggestions, We added this information according to OECD guidelines; OECD 407-2008 and 420-2001 which it was referenced.

Rationale of animal used: OECD 407-2008; At least 10 animals (five female and five male) should be used at each dose level.

Diet information was cleared in the method section.

Effectively, we used  a dose of 1000 mg/Kg BW in our study Generally, at least three test groups and a control group should be used, but if from assessment of other data, no effects would be expected at a dose of 1000mg/kg bw/d, a limit test may be performed.(https://ntp.niehs.nih.gov/iccvam/suppdocs/feddocs/oecd/oecdtg407-2008.pdf)

3. Out of normal range in hematological parameters such as hematocrit and hemoglobin.

R4. Thank you for your suggestions, We corrected both tables in biochemical and hematological because we divided those values ​​by the body weight of each animal and multiply by 100 as in relative organ weights. I beg you accept my apologies. 

Reviewer 2 Report

This is a generally solid basic paper investigating the potential toxicity of potato tocosh, a traditional Peruvian processed potato. Guideline studies were used to investigate the acute and short-term toxicity. Considering the toxicity is of interest due to the known potential for potato to contain alkaloids at toxic levels. However, the utility and broader applicability of the paper is limited by the limited phytochemical characterization.

The authors do a nice job of characterizing the collection of the tocosh (is it the tocosh or the potatoes that are collected?). And they note that a phytochemical analysis was conducted of the tocosh. Was the analysis only qualitative? Is it possible to obtain information on the concentration of key alkaloids in the flour that was used for the study? This would allow the determination of the dose of these alkaloids in the toxicity tests. Without this information, we only have the data on one specific batch of tocosh, but if there were some quantitative data, it would be easier to generalize the results to other samples.

The paper will be easier to follow if it is re-ordered in a more traditional organization. First present all of the results of the acute single-dose study, then all of the results of the 28-day study. Also, since one study is in mice and the other in rats, it would be useful to specify the species more frequently.

With regard to the toxic findings, LDL showed a significant increase in female rats. Although not statistically significant, it is noteworthy that HDL was decreased in the female rats. Similarly, there was a non-significant increase in LDL and decrease in HDL in males. These changes suggest a pattern that is worthy of further investigation, particularly in light of the increased body weight relative to controls. Although decreased body weight is usually the concern in toxicity studies, this pattern of changes could reflect disregulation, or may simply reflect the high carbohydrate content of the tocosh flour.

Line 65: Please provide a reference regarding natural penicillin.

It would be useful if more standard histopathology terms are used.

How were effects of cholinesterase inhibition evaluated, aside from monitoring clinical signs of toxicity?

Line 340: please provide the doses in mg/kg for ease of comparison with the current study.

Lines 360-366: Please be clear on which studies were in vivo and which in vitro

Aside from general editing for improved clarity and proper English, there are some statements where the meaning is unclear. A few examples:

Lines 56-57: “At the end of the process, potato is reduced in size, except its peel and 56 get a very peculiar unpleasant odor.”

Lines 132-133: “Mice were maintained with water but not with food that was retained for 3 to 4 hours, the process was continued on an empty stomach.”

Line 229: “evident toxicity and/or 1 ≤ death” – should this be “≤ 1 death”?

Line 382: “tocosh could be consumed in doses stablished until 1000 mg/Kg.”

Author Response

Dear Reviewer, thank you for your observations and suggestions with our article. We highlighted with purple and yellow color some modifications.

This is a generally solid basic paper investigating the potential toxicity of potato tocosh, a traditional Peruvian processed potato. Guideline studies were used to investigate the acute and short-term toxicity. Considering the toxicity is of interest due to the known potential for potato to contain alkaloids at toxic levels. However, the utility and broader applicability of the paper is limited by the limited phytochemical characterization.

The authors do a nice job of characterizing the collection of the tocosh (is it the tocosh or the potatoes that are collected?). And they note that a phytochemical analysis was conducted of the tocosh. Was the analysis only qualitative? Is it possible to obtain information on the concentration of key alkaloids in the flour that was used for the study? This would allow the determination of the dose of these alkaloids in the toxicity tests. Without this information, we only have the data on one specific batch of tocosh, but if there were some quantitative data, it would be easier to generalize the results to other samples.

R1.- Dear reviewer, thank you for your comments, in fact we did not isolate any molecule of potato tocosh, just qualitative phytochemical analysis. However, tocosh is made from Solanum tuberosum tubers and steroidal alkaloids have been characterized such as solanine, solanidine, chaconine. Furthermore, they represent the majority of their phytochemicals.

In regard to Tocosh flour, we did not find literatures about tocosh only 3 papers which they were cited in this manuscript.

  1. Jiménez E, Yépez A, Pérez-Cataluña A, Ramos Vasquez E, Zúñiga Dávila D, Vignolo G, et al. Exploring diversity and biotechnological potential of lactic acid bacteria from tocosh - Traditional Peruvian fermented potatoes - By high throughput sequencing (HTS) and culturing, LWT - Food Science and Technology, 2018, 109:168-74.
  2. Mayta-Tovalino F, Sedano-Balbin G, Romero-Tapia P, et al. Development of New Experimental Dentifrice of Peruvian Solanum tuberosum (Tocosh) Fermented by Water Stress: Antibacterial and Cytotoxic Activity. J Contemp Dent Pract. 2019;20(10):1206–1211.
  3. Mosso AL, Jimenez ME, Vignolo G, LeBlanc JG, Samman NC. Increasing the folate content of tuber based foods using potentiasally probiotic lactic acid bacteria. Food Res Int. 2018; 109:168–174.

Those articles do not mention any quantification or chemical analysis to extrapolate with our results. The interest of those articles is based only on their probiotic content. We are doing another project to characterize those molecules that probably were generated during fermentation process.

Figure 1 shows how people obtain Tocosh, which represents extraction after 6 months of fermentation under a water well.

The paper will be easier to follow if it is re-ordered in a more traditional organization. First present all of the results of the acute single-dose study, then all of the results of the 28-day study. Also, since one study is in mice and the other in rats, it would be useful to specify the species more frequently.

R2.- Dear reviewer, thank you for your comments, we ordered according to your suggestions.

With regard to the toxic findings, LDL showed a significant increase in female rats. Although not statistically significant, it is noteworthy that HDL was decreased in the female rats. Similarly, there was a non-significant increase in LDL and decrease in HDL in males. These changes suggest a pattern that is worthy of further investigation, particularly in light of the increased body weight relative to controls. Although decreased body weight is usually the concern in toxicity studies, this pattern of changes could reflect disregulation, or may simply reflect the high carbohydrate content of the tocosh flour.

R3. In fact, this variation could be the effect of high content of carbohydrates in potato flour and we mentioned in discussion.

Line 65: Please provide a reference regarding natural penicillin.

R3.- Dear reviewer, thank you for your comments, we added that reference.

It would be useful if more standard histopathology terms are used.

R4.- Dear reviewer, thank you for your comments, we modified according to your suggestions.

How were effects of cholinesterase inhibition evaluated, aside from monitoring clinical signs of toxicity?

  R5 They were monitored by using this table:

General Behavior

Control Group

Treatment

1000 mg/kg b.w.

Decrease of motor activity

Not present

Not present

Increase of motor activity

Not present

Not present

Loss of reflections or straightening

Not present

Not present

Change in the skin

Not present

Not present

Tail erection

Not present

Not present

Piloerection

Not present

Not present

Drowsiness

Not present

Not present

Diarrhea

Not present

Not present

Aggressive

Not present

Not present

Afraid

Not present

Not present

Death

Alive

Alive

Weight variation

Not present

present

To evidence any effect on cholinesterase inhibition, Excess secretions and muscular twitching were absent as well as frequency of feces, breathing and dilated pupils.

Line 340: please provide the doses in mg/kg for ease of comparison with the current study.

R6.- Dear reviewer, thank you for your comments, we modified according to your suggestions.

 Lines 360-366: Please be clear on which studies were in vivo and which in vitro

R7.- Dear reviewer, thank you for your comments, we modified according to your suggestions.

Aside from general editing for improved clarity and proper English, there are some statements where the meaning is unclear. A few examples:

R8.- Dear reviewer, thank you for your comments, we modified according to your suggestions.

Lines 56-57: “At the end of the process, potato is reduced in size, except its peel and 56 get a very peculiar unpleasant odor.”

R9.- Dear reviewer, thank you for your comments, we modified according to your suggestions.

Lines 132-133: “Mice were maintained with water but not with food that was retained for 3 to 4 hours, the process was continued on an empty stomach.”

R10.- Dear reviewer, thank you for your comments, we modified according to your suggestions.

Line 229: “evident toxicity and/or 1 ≤ death” – should this be “≤ 1 death”?

R11.- Dear reviewer, thank you for your comments, we modified according to your suggestions.

Line 382: “tocosh could be consumed in doses stablished until 1000 mg/Kg.”

R12.- Dear reviewer, thank you for your comments, we modified according to your suggestions.

Reviewer 3 Report

My

General Comments

  • Always present all of the study first in methods, results, discussion.
  • Follow with subacute study next in methods, results, discussion. Do not co-mingle the two studies.
  • Results sections should be separated into those for the acute study in mice and those for the subacute study for rats.
  • Add, if possible, the estimated daily mg/kg TF consumed per person per day mg/kg/day from the diet. Add, concentrations of the various alkaloid constituents in the same units through out, either ug/g TF or mg/kg TF. From this information, estimate the total alkaloid exposure to humans on a daily basis in mg alkaloids/kg BW.
  • Why was histopathology done in the acute study? The OECD Guideline 420 only requires histopathology IF there is gross pathology: “All gross pathological changes should be recorded for each animal. Microscopic examination of organs showing evidence of gross pathology in animals surviving 24 or more hours after the initial dosing may also be considered because it may yield useful information.” The authors do not indicate any gross pathology findings. These findings if present or if absent, should be recorded. Control mice should have been considered to compare to the pathology lesions in Figures 4 and 5. It is unknown if the organ histopathology is treatment-related. This verification of toxicity presents a concern that is not discussed. It should be fully discussed in the Discussion section.

Detailed Comments

Page 4:

Section 2.5 the numbering of the sections should only include the Acute Oral Toxicity. 2.5.1 Sub-acute oral toxicity should be a separate section, i.e. 2.6. The methods for each study, either acute or subacute, should be separated into its own section to avoid confusion.

Section 2.5.1: Correction to line 117: Explain “and the empirical information for consumption of tocosh by the population”. What is the human ingestion of tocosh on a daily mg/kg basis? This information should be added to the paper.

Section 2.5.2 Line 128-131: Question again on the amount of tocosh consumed daily by humans, add exact data. Line 141: The word “notoriety” should be changed to “toxicity.”

Section 2.52 Line 141-145: It is not clear whether the organs were examined by gross pathology as no gross pathology results are included to justify a microscopic evaluation of the organs. Since that is missing, either add details to the methods to justify histopathology or discuss the findings in results, see below.

Page 6

Section 3.1, line 199: Were the percentages of the various alkaloids and constituents analyzed? Give the results and include quantities or percentages.

Table 1: Liver in tocosh group males and females was statistically increased compared to control. The authors should mention and discuss this finding. Perhaps, historical control data on the rat species used could be checked and included for comparison to see if the statistical difference is biologically within normal variability or treatment-related. Discuss in the Discussion

Page 7

Table 2: Check statistical findings in both sexes and compare them to historical control animals for interpretation, as in comment for Table 1.

Page 9 & Page 10: Figures 4 and 5

Section 3.3: Line 228-230: OECD Guideline 420-2001 states, “All gross pathological changes should be recorded for each animal. Microscopic examination of organs showing evidence of gross pathology in animals surviving 24 or more hours after the initial dosing may also be considered because it may yield useful information.” The authors did not state that they examined the organs on a gross scale. Only if they found changes on gross pathology exam should they have done the histopathology. No control animals in the laboratory were checked for the liver and kidney lesions. It is possible the lesions could have been due to an infection or other problem in the laboratory, but without data from control animals, it is not possible to determine.

Page 11

Section 3.4, line 292-294: Remove body weight information on mice from the section. Include rat and mouse studies in separate section through out the Results!

Page 12 & 13

Section 4 Discussion: Line 328: give units for glucoalkaloid content in units of mg/kg so they can be compared directly with units on Line 353, 355, 358. Confusing

Lines : 348-351: Sentence is not clear. Explain more clearly.

Lines 352-359: Paragraph seems to confuse the toxicity of the direct alkaloids with the TOTE Tox data: Explain and clarily.

Line 382-383: The data do not support consumption at 1000 mg/kg. Typically, the No Observed Effect Level (NOEL) from an animal study is divided by a 100-fold safety factor to determine safety to humans, i.e. 1000 mg/kg divided by 100 = 10 mg/kg shown to be safe for human consumption! This risk assessment procedure is standard throughout the world. Therefore, the known human daily consumption of tocosh is critically important to understand the results of this manuscript and should be used to put the findings into perspective.

                        a

Author Response

Dear reviewer thanks for your recommendations with our paper. Modifications were highlighted with yellow color.

General Comments

  • Always present all of the study first in methods, results, discussion.
  • Follow with subacute study next in methods, results, discussion. Do not co-mingle the two studies.
  • Results sections should be separated into those for the acute study in mice and those for the subacute study for rats.
  • Add, if possible, the estimated daily mg/kg TF consumed per person per day mg/kg/day from the diet. Add, concentrations of the various alkaloid constituents in the same units throughout, either ug/g TF or mg/kg TF. From this information, estimate the total alkaloid exposure to humans on a daily basis in mg alkaloids/kg BW.
  • Why was histopathology done in the acute study? The OECD Guideline 420 only requires histopathology IF there is gross pathology: “All gross pathological changes should be recorded for each animal. Microscopic examination of organs showing evidence of gross pathology in animals surviving 24 or more hours after the initial dosing may also be considered because it may yield useful information.” The authors do not indicate any gross pathology findings. These findings if present or if absent, should be recorded.  Control mice should have been considered to compare to the pathology lesions in Figures 4 and 5.  It is unknown if the organ histopathology is treatment-related. This verification of toxicity presents a concern that is not discussed. It should be fully discussed in the Discussion section.

Detailed Comments

Page 4:

Section 2.5 the numbering of the sections should only include the Acute Oral Toxicity.  2.5.1 Sub-acute oral toxicity should be a separate section, i.e. 2.6.  The methods for each study, either acute or subacute, should be separated into its own section to avoid confusion.

R1.- Dear reviewer, thank you for your comments, we modified according to your suggestions.

Section 2.5.1: Correction to line 117: Explain “and the empirical information for consumption of tocosh by the population”. What is the human ingestion of tocosh on a daily mg/kg basis? This information should be added to the paper.

R2.- Dear reviewer, thank you for your comments, although, tocosh flour consumption is invariable, the normal dose known in the traditional medicine is between 500 and 1000 mg/Kg daily, divided three times a day. Generally, it is for gastric disorders.

Section 2.5.2 Line 128-131: Question again on the amount of tocosh consumed daily by humans, add exact data.  Line 141: The word “notoriety” should be changed to “toxicity.”

R3.- Dear reviewer, thank you for your comments, we modified according to your suggestions.

Data of tocosh consumed daily by humans was included in the introduction.

Section 2.52 Line 141-145: It is not clear whether the organs were examined by gross pathology as no gross pathology results are included to justify a microscopic evaluation of the organs.  Since that is missing, either add details to the methods to justify histopathology or discuss the findings in results, see below.

R4.- We evaluated all organs in mice and rats and they were included in supplementary material. We included in the manuscript figures of lesions of liver and kidney.

Page 6

Section 3.1, line 199: Were the percentages of the various alkaloids and constituents analyzed? Give the results and include quantities or percentages.

R5.- Dear reviewer, thank you for your comments, in fact we did not isolate any molecule of potato tocosh, just qualitative phytochemical analysis. However, tocosh is made from Solanum tuberosum tubers and steroidal alkaloids have been characterized such as solanine, solanidine, chaconine. Furthermore, they represent the majority of their phytochemicals.

In regard to Tocosh flour, we did not find literatures about tocosh only 3 papers which they were cited in this manuscript.

  1. Jiménez E, Yépez A, Pérez-Cataluña A, Ramos Vasquez E, Zúñiga Dávila D, Vignolo G, et al. Exploring diversity and biotechnological potential of lactic acid bacteria from tocosh - Traditional Peruvian fermented potatoes - By high throughput sequencing (HTS) and culturing, LWT - Food Science and Technology, 2018, 109:168-74.
  2. Mayta-Tovalino F, Sedano-Balbin G, Romero-Tapia P, et al. Development of New Experimental Dentifrice of Peruvian Solanum tuberosum (Tocosh) Fermented by Water Stress: Antibacterial and Cytotoxic Activity. J Contemp Dent Pract. 2019;20(10):1206–1211.
  3. Mosso AL, Jimenez ME, Vignolo G, LeBlanc JG, Samman NC. Increasing the folate content of tuber based foods using potentiasally probiotic lactic acid bacteria. Food Res Int. 2018; 109:168–174.

Those articles do not mention any quantification or chemical analysis to extrapolate with our results. The interest of those articles is based only on their probiotic content. We are doing another project to characterize those molecules that probably were generated during fermentation process.

Table 1: Liver in tocosh group males and females was statistically increased compared to control. The authors should mention and discuss this finding. Perhaps, historical control data on the rat species used could be checked and included for comparison to see if the statistical difference is biologically within normal variability or treatment-related. Discuss in the Discussion.

R6.- Dear reviewer, thank you for your comments, we modified according to your suggestions. Line 460-466, pp. 18

Page 7

Table 2: Check statistical findings in both sexes and compare them to historical control animals for interpretation, as in comment for Table 1.

R7.- Dear reviewer, thank you for your comments, we modified according to your suggestions.

Page 9 & Page 10: Figures 4 and 5

Section 3.3: Line 228-230: OECD Guideline 420-2001 states, “All gross pathological changes should be recorded for each animal. Microscopic examination of organs showing evidence of gross pathology in animals surviving 24 or more hours after the initial dosing may also be considered because it may yield useful information.”  The authors did not state that they examined the organs on a gross scale.  Only if they found changes on gross pathology exam should they have done the histopathology.  No control animals in the laboratory were checked for the liver and kidney lesions.  It is possible the lesions could have been due to an infection or other problem in the laboratory, but without data from control animals, it is not possible to determine.

R8. In fact, we do not use a control group because the OECD guideline consider 5 animals per group in limit test. We evaluated all organs in mice and they were included in supplementary material. We could think that those lesions were by infections. However, rats and mice were in the same conditions and similar environment during the evaluation. We also included images of rat organs of control and treated group and did not evidenced any lesions. (see supplementary material).

Page 11

Section 3.4, line 292-294: Remove body weight information on mice from the section. Include rat and mouse studies in separate section throughout the Results!

R9.- Dear reviewer, thank you for your comments, we modified according to your suggestions.

Page 12 & 13

Section 4 Discussion: Line 328: give units for glucoalkaloid content in units of mg/kg so they can be compared directly with units on Line 353, 355, 358. Confusing

R10.- Dear reviewer, thank you for your comments, we modified according to your suggestions.

Lines : 348-351: Sentence is not clear. Explain more clearly.

R11.- Dear reviewer, thank you for your comments, we modified according to your suggestions.

Lines 352-359: Paragraph seems to confuse the toxicity of the direct alkaloids with the TOTE Tox data: Explain and clarily.

R12.- Dear reviewer, thank you for your comments, we cleared that paragraph.

Line 382-383: The data do not support consumption at 1000 mg/kg. Typically, the No Observed Effect Level (NOEL) from an animal study is divided by a 100-fold safety factor to determine safety to humans, i.e. 1000 mg/kg divided by 100 = 10 mg/kg shown to be safe for human consumption! This risk assessment procedure is standard throughout the world.  Therefore, the known human daily consumption of tocosh is critically important to understand the results of this manuscript and should be used to put the findings into perspective.

R13. Dear reviewer, The NOAEL in rats of tocosh flour was 1000 mg/Kg, then to estimate the human equivalent dose (HED) (divided by 0.162) corresponds a dose of 162 mg/Kg, which in an adult of 60 Kg and divided by a factor of 10 gives 1000 mg as the initial dose in humans.

https://www.ncbi.nlm.nih.gov/pmc/articles/PMC4804402/

Round 2

Reviewer 1 Report

There are major points to be addressed

- acute toxicity was not performed in according to OECD 420 guideline. Authors used both sexes of mice with fasting time of 4-6 hr, instead of one sex of rats (female) with 3-4 hr. So the description with a compliance with OECD 420 guideline should be deleted from page 3, 16 and 22 of the manuscript.

- lack of some information on age of mice and rats used, dosing volume of test substance and dosing route of sodium pentobarbital.

- page 11, line 293 to 294. it was decribed there was no significance in relative organ weights. But in Table 1, relative liver weights increased with statistical significance in both sexes treated with 1000 mg/kg

- page 11, line 304 to 305, the sentence “only a slight variation in the significant increase in LDL cholesterol was recorded in rats. female and AST in male rats. (Table2)” should be changed to “only a slight variation in the significant increase in LDL cholesterol was recorded in rats and decreased triglycerides, in male rats. (Table 2).”

- page 11, line 313, the statistical value in body weights of females rats is changed from P<0.05 to p<0.001.

- page 14, Table 3, there was no statistical mark in monocyte percentage in male rats given 1000 mg/kg

- page 16, line 401. a typo, change testicle to testes

- page 21, line 534, there is an inconsistency in the description of decreased HDL among results section, Table 2, and discussion section.

Author Response

Dear Reviewer 

Thank you for your recommendations and suggestions.

Acute toxicity was not performed in according to OECD 420 guideline. Authors used both sexes of mice with fasting time of 4-6 hr, instead of one sex of rats (female) with 3-4 hr. So the description with a compliance with OECD 420 guideline should be deleted from page 3, 16 and 22 of the manuscript.

R1.- Dear reviewer, you have reason with your commentaries. However, The OECD 420 states: The preferred rodent species is the rat, although other rodent species may be used. Normally females are used. This is because literature surveys of conventional LD50 tests show that usually there is little difference in sensitivity between the sexes, but in those cases where differences are observed, females are generally slightly more sensitive (10). However, if knowledge of the toxicological or toxicokinetic properties of structurally related chemicals indicates that males are likely to be more sensitive then this sex should be used. When the test is conducted in males, adequate justification should be provided.

We are going to add the term with slight modifications in animal selection, sex and fasting.

Furthermore; we found a study by

Park S, Park MY, Song G, Lim W. Alpha-solanine inhibits cell proliferation via mitochondrial dysfunction and inhibin synthesis in mouse testis In vitro and In vivo. Chemosphere. 2019;235:271‐279. doi:10.1016/j.chemosphere.2019.06.172

Which, they reported that solanine produced toxic effects on testes and male reproductive function in mice. However, in our study we did not evidence any alterations in sexual organs.

lack of some information on age of mice and rats used, dosing volume of test substance and dosing route of sodium pentobarbital.

R2.-We added those informations. Adult healthy male and female Holztman albino rats (age, 12 weeks: body weight (males), 160–180 g; body weight (females), 150–170 g) were used to evaluate the sub-acute toxicity. Male and female rats were housed separately, and the selected female rats were nulliparous and non-pregnant. In the acute toxicity, adult healthy male and female albino mice (age, 8 weeks: body weight (males), 30–32 g; body weight (females), 25–30 g) were used during the evaluation. 

page 11, line 293 to 294. it was decribed there was no significance in relative organ weights. But in Table 1, relative liver weights increased with statistical significance in both sexes treated with 1000 mg/kg

R3.- Dear reviewer, this paragraph was changed according to results of table 1.

page 11, line 304 to 305, the sentence “only a slight variation in the significant increase in LDL cholesterol was recorded in rats. female and AST in male rats. (Table2)” should be changed to “only a slight variation in the significant increase in LDL cholesterol was recorded in rats and decreased triglycerides, in male rats. (Table 2).”

R4.- Dear reviewer, this paragraph was changed.

page 11, line 313, the statistical value in body weights of females rats is changed from P<0.05 to p<0.001.

R5.- Dear reviewer, this paragraph was modified.

page 14, Table 3, there was no statistical mark in monocyte percentage in male rats given 1000 mg/kg

R6.- Dear reviewer, we added the statistical mark.

page 16, line 401. a typo, change testicle to testes

R7.- Dear reviewer, we changed to testes.

page 21, line 534, there is an inconsistency in the description of decreased HDL among results section, Table 2, and discussion section.

R8.- Dear reviewer, this paragraph was modified.

Reviewer 3 Report

 Comments on Revised Manuscript

Page 2, line 68: change to “with high safety at up to 1000 mg/kg BW for 28 days in rats.” This is a more accurate statement.

Page 3, line 89-90: Provide a reference for the statement of human consumption at 500-1000 mg/kg BW.

Page 7, line 183: ADD, were the organs examined for gross and/or microscopic pathology? ADD.

Page 8, line 231, section 2.7 Histopathological Analysis: ADD from which study were the tissues evaluated, acute and/or subacute? ADD: include a list of organs examined for histopathology. Needs clarification.

Pages 11-13, Tables 1, 2 and 3: ADD the historical control values for those parameters which showed statistical significance for the dosed groups. If not available, state that no historical control values are available. Note, the laboratory supplying the rats should have this information.

Page 16, line 397: ADD, note if any gross pathology lesions were noted in the treated animals. The methods sections should state if the organs were examined grossly for abnormal lesions or not examined.

Page 21, line 529: ADD to final sentence, “This research revealed that presence of any phytochemical groups found in tocosh flour 529 did not alter the histology in rats at the repeated dose of 1000 mg/kg BW for 28 days.”

Page 22, line 538, ADD, “No historical control data on the clinical chemistry or hematology values from the animal supplier were available for comparison; therefore, it is possible that the statistically changed parameters noted were within the normal range of average parameter values.”

Page 23, lines 576-580, Re-write the following as follows sentences, “The NOAEL in rats of tocosh flour was 1000 mg/Kg, then to estimate the human equivalent dose 577 (HED) [39] corresponds a dose of 1000 mg as initial dose in humans with 60 Kg of body weight. 578 According to the traditional consumption of tocosh, this dose is less than the dose consumed by 579 the population. Although, it seems to be safe compared with the limit dose of glycoalkaloids found 580 in potato tubers.”

What is the HED and how is it calculated? Explain the meaning of the statements. The NOAEL of 1000 mg/kg BW applied to humans of 60 kg BW would be 1000 mg/kg x 60 = 60,000 mg total dose. Earlier in the manuscript, it is stated that humans consume up to 1000 mg/kg BW. How does the HED compare to the estimated amount humans consume. This is important and needs to be explained better. It seems that the NOAEL dose is the human consumption, leaving no safety factor or room for increase.

Page 23, line 584, “Future studies of 585 genotoxicity and chronical toxicity are needed as well as standardize its dose for any ailment.” Correct, “Future studies of 585 genotoxicity and chronic toxicity are needed as well as standardize its dose for consumption in foods.”

Page 23, line 588, “In the acute oral toxicity of a fixed-dose procedure, at the doses 589 of 2000 and 5000 mg/kg BW was evidenced minor histopathological changes in liver and kidney 590 structures but keep unclear these findings.” Change the statement “There were no deaths at up to 5000 mg/kg bw, therefore, the oral LD50 as greater than 5000 mg/kg.”

Author Response

Dear reviewer, thank you for your observations and suggestions.

Page 2, line 68: change to “with high safety at up to 1000 mg/kg BW for 28 days in rats.” This is a more accurate statement.

R1.-Dear reviewer, thank you, we modified according to your suggestion.

Page 3, line 89-90: Provide a reference for the statement of human consumption at 500-1000 mg/kg BW.

R2.-Dear reviewer, thank you, we have no reference on the dose, we stated in the introduction (this information was taken according to the interview at the place where potato tocosh was collected). This manuscript will be the first report on tocosh dose.

Page 7, line 183: ADD, were the organs examined for gross and/or microscopic pathology? ADD.

R3.-Dear reviewer, thank you, we added that sentence.

Page 8, line 231, section 2.7 Histopathological Analysis: ADD from which study were the tissues evaluated, acute and/or subacute? ADD: include a list of organs examined for histopathology. Needs clarification.

R3.-Dear reviewer, thank you, we added that sentence. Brain, heart, lung, liver, spleen. stomach, kidney, testes and uterus of both studies (sub-acute and acute toxicity) .....

Pages 11-13, Tables 1, 2 and 3: ADD the historical control values for those parameters which showed statistical significance for the dosed groups. If not available, state that no historical control values are available. Note, the laboratory supplying the rats should have this information.

R4.- Thank you dear, we added, "No historical control values ​​available" in each table.

Page 16, line 397: ADD, note if any gross pathology lesions were noted in the treated animals. The methods sections should state if the organs were examined grossly for abnormal lesions or not examined.

R5.-Dear reviewer, thank you, we added "organs were examined grossly for abnormal lesions" in section method in both studies. 

We added in results: Mice administered at doses of 2000 mg/Kg and 5000 mg/kg body weight exhibited minor organ damage in liver (parenchymal lymphocyte) and kidney (lymphocyte) according to Fig. 4 and Fig. 5.  However, no external aspects of toxicity were observed during the study linked to liver and kidney damage.

Page 21, line 529: ADD to final sentence, “This research revealed that presence of any phytochemical groups found in tocosh flour 529 did not alter the histology in rats at the repeated dose of 1000 mg/kg BW for 28 days.”

R6.-Dear reviewer, thank you, we added according to your suggestions.

Page 22, line 538, ADD, “No historical control data on the clinical chemistry or hematology values from the animal supplier were available for comparison; therefore, it is possible that the statistically changed parameters noted were within the normal range of average parameter values.”

R7.-Dear reviewer, thank you, we added according to your suggestions.

Page 23, lines 576-580, Re-write the following as follows sentences, “The NOAEL in rats of tocosh flour was 1000 mg/Kg, then to estimate the human equivalent dose 577 (HED) [39] corresponds a dose of 1000 mg as initial dose in humans with 60 Kg of body weight. 578 According to the traditional consumption of tocosh, this dose is less than the dose consumed by 579 the population. Although, it seems to be safe compared with the limit dose of glycoalkaloids found 580 in potato tubers.”

R8.- Dear reviewer, thank you, we added according to your suggestions.

What is the HED and how is it calculated? Explain the meaning of the statements. The NOAEL of 1000 mg/kg BW applied to humans of 60 kg BW would be 1000 mg/kg x 60 = 60,000 mg total dose. Earlier in the manuscript, it is stated that humans consume up to 1000 mg/kg BW. How does the HED compare to the estimated amount humans consume. This is important and needs to be explained better. It seems that the NOAEL dose is the human consumption, leaving no safety factor or room for increase.

R9.-The dose by factor method is an empirical approach and use the no observed adverse effect levels (NOAEL) of drug from preclinical toxicological studies to estimate human equivalent dose (HED).

https://www.ncbi.nlm.nih.gov/pmc/articles/PMC4804402/

HED = animal dose in mg/kg x (animal weight in kg/human weight in kg)0.33

the factor for rats is 0.162.

HED = 1000 mg/kg X 0.162 =162 mg/kg, Thus, for a 60 kg human, the dose is 9720 mg (162 X 60). This HED value is further divided by a factor value of 10 (safety factor); thus, the initial dose in entry into man studies is 972 mg, approaching to 1000 mg as the total dose.

Page 23, line 584, “Future studies of 585 genotoxicity and chronical toxicity are needed as well as standardize its dose for any ailment.” Correct, “Future studies of 585 genotoxicity and chronic toxicity are needed as well as standardize its dose for consumption in foods.”

R10.- Dear reviewer, thank you, we changed it according to your suggestions.

Page 23, line 588, “In the acute oral toxicity of a fixed-dose procedure, at the doses 589 of 2000 and 5000 mg/kg BW was evidenced minor histopathological changes in liver and kidney 590 structures but keep unclear these findings.” Change the statement “There were no deaths at up to 5000 mg/kg bw, therefore, the oral LD50 as greater than 5000 mg/kg.”

R11.- Dear reviewer, thank you, we changed it according to your suggestions.